# The Design and Development of Recycled Concretes in a Circular Economy Using Mixed Construction and Demolition Waste

**DOI:** 10.3390/ma14164762

**Published:** 2021-08-23

**Authors:** Marcos Díaz González, Pablo Plaza Caballero, David Blanco Fernández, Manuel Miguel Jordán Vidal, Isabel Fuencisla Sáez del Bosque, César Medina Martínez

**Affiliations:** 1Department of Construction Sciences, Metropolitan Technological University, Dieciocho, Santiago de Chile 161, Chile; dblanco@utem.cl; 2Department of Construction, Research Institute for Sustainable Territorial Development (INTERRA), University of Extremadura, 10003 Cáceres, Spain; pcaballerop@unex.es (P.P.C.); isaezdelu@unex.es (I.F.S.d.B.); cmedinam@unex.es (C.M.M.); 3Department of Agrochemistry and Environment, Miguel Hernández University of Elche, 03202 Elche, Spain; manuel.jordan@umh.es

**Keywords:** recycled concrete, recycled mixed sand, recycled mixed gravel, mechanical properties, strength, watertightness

## Abstract

This research study analysed the effect of adding fine—fMRA (0.25% and 50%)—and coarse—cMRA (0%, 25% and 50%)—mixed recycled aggregate both individually and simultaneously in the development of sustainable recycled concretes that require a lower consumption of natural resources. For this purpose, we first conducted a physical and mechanical characterisation of the new recycled raw materials and then analysed the effect of its addition on fresh and hardened new concretes. The results highlight that the addition of fMRA and/or cMRA does not cause a loss of workability in the new concrete but does increase the amount of entrained air. Regarding compressive strength, we observed that fMRA and/or cMRA cause a maximum increase of +12.4% compared with conventional concrete. Tensile strength increases with the addition of fMRA (between 8.7% and 5.5%) and decreases with the use of either cMRA or fMRA + cMRA (between 4.6% and 7%). The addition of fMRA mitigates the adverse effect that using cMRA has on tensile strength. Regarding watertightness, all designed concretes have a structure that is impermeable to water. Lastly, the results show the feasibility of using these concretes to design elements with a characteristic strength of 25 MPa and that the optimal percentage of fMRA replacement is 25%.

## 1. Introduction

Climate change and global warming have become important issues that directly impact the world economy. This has led to countries developing laws and regulations that control the emission of CO_2_ and the appropriate management of waste. In this context, the construction industry is responsible for 12% of all greenhouse gas emissions in the European Union (EU) and for generating ~25–30% of the solid waste produced every year in the EU [1], which equates to an average annual production of 800 million tonnes. In addition, as many as 534 and 200 million tonnes of construction and demolition waste (C&DW) are generated every year in the United States and China, respectively.

Concrete is the most used material in the construction sector worldwide, with the EU and USA producing 165 and 150 million cubic metres, respectively, in 2018 [2]. This industry is characterised by requiring large amounts of natural resources, as this material is mainly composed of aggregates—60–75% of the volume of concrete—and, to a lesser extent, cement—10–15%. Its manufacturing process is responsible for ~8% of the total worldwide amount of CO_2_ [3]. Three trillion tonnes of aggregate were used in the field of civil engineering in 2018, ~45% of which were used to manufacture concrete [4].

In this context, the construction industry has tried to mitigate the adverse effects of its activity in recent years by designing and developing new formulations for materials with a cement base (mortars and/or concretes) that are more sustainable [5,6]. This has been done by adding industrial by-products (agroforestry or biomass waste, ornament and ceramics [7] industry and construction and demolition waste [8,9]) as supplementary cementitious materials and/or recycled aggregate (concrete or mixed waste from C&DW). The use of recycled aggregate from C&DW is one of the most widespread strategies for simultaneously achieving the double goal of instituting the concept of circular economy and sustainability in construction. 

Recycled aggregates are obtained from the appropriate treatment of C&DW in management plants, where they are classified in granulometric fractions (mainly sand, coarse and gravel) depending on their size and composition. According to this last criterion, they are classified as (i) recycled concrete aggregates (RCA) with amounts of concrete (Rc) and unbound aggregates (Ru) ≥90% and ≤10% of ceramic materials (Rb), (ii) mixed recycled aggregates (MRA) with 70% ≤ Rc + Ru < 90% and Rb ≤ 30% and (iii) recycled masonry aggregates (RMA) with Rc + Ru < 70% and Rb > 30.

RCA are recycled aggregates that have been studied in greater depth in the international literature, focusing mainly on the coarse fraction (maximum particle size ≥ 4 mm). In this line of work, it is worth highlighting the study conducted by Bravo et al. [10], who found that the density decreased between 4.7% and 7.7% by adding 100% of RCA, while considering that their mechanical strength decreases if their composition includes particles of a ceramic nature. Moreover, Thomas et al. [11] concluded that the density of concretes decreased by 5% with a 20% of RCA replacement due to the high porosity and the presence of adhered mortar. Etxeberría et al. [12] analysed concretes with 100% RCA, a water–cement ratio of 0.5 and 325 kg of cement/m^3^, registering a decrease between 20% and 25% decrease in compressive strength compared with the reference concrete. This loss could be offset by increasing the cement content, a strategy which is of no interest economically or from the viewpoint of sustainability. Likewise, these authors observed that concretes with average compressive strength (30–45 MPa) manufactured with 25% recycled coarse aggregate had the same mechanical properties as traditional concrete. McNeil et al. [13], in their literature review, were able to summarise their findings on what concretes with RCA experience: (1) replacing natural aggregate (NA) in concrete with RCA decreases the compressive strength but yields comparable splitting tensile strength; (2) the modulus of rupture for RCA concrete was slightly lower than that of conventional concrete, likely due to the weakened interfacial transition zone (ITZ) from residual mortar; and (3) the elastic modulus is also lower than expected, caused by the more ductile aggregate.

As regards assessing the fine fraction of RCA, there have been fewer studies that have focused on studying the design of structural concretes. Evangelista et al. [14] assessed the feasibility of adding the fine fraction of RCA and observed that the compressive strength was not impacted for percentages ≤30% in weight. Minkwan et al. [15] found that the compressive strength of concrete with 100% of fine RCA decreased by approximately 30% and 10% compared with the reference concrete with normal strength and high strength, respectively. Bravo et al. [16] observed that (i) recycled mixes with contents of fine RCA ≤25% have properties that are comparable with those of the reference concrete and (ii) contents of RCA >25% cause decreases in the properties, leading to losses of up to 19% compared with traditional concrete. 

Regarding the simultaneous use of coarse and fine RCA, Fernández et al. [17] concluded that the properties that are impacted the most due to replacing natural aggregate with recycled aggregate are workability (with 100% coarse aggregate replacement, it more than doubles compared with the reference concrete), the elastic modulus (decreases by 42% with 100% coarse aggregate replacement and also decreases by 7% with 30% fine aggregate replacement), the contractive deformation (obtaining deformations between 40% and 56% after 360 days with 100% coarse aggregate replacement) and water absorption (between 8.5% and 9%, proportions that are higher than the 5% established by the Spanish Code on Structural Concrete, or Structural Concrete Instruction EHE-08). 

Plaza et al. [18] report that the use of small percentages (<25%) of coarse concrete aggregate alone or with recycled concrete sand decreases the total porosity and refines the structure of the pores. The opposite effect was observed for higher percentages. Recycled concretes have less compressive strength after 28 days than the conventional material, although the decrease was less than 17% in all cases studied. The loss of strength is greater in mixes that have recycled mixed fines. Regarding tensile strength, Plaza et al. [18] revealed that it increased by 9.49% with a coarse recycled aggregate replacement rate of 100%. However, in replacement mixes (50% coarse aggregate + 50% fine aggregate) this property decreased by up to 14.13%. Furthermore, there was a 1.74% increase in strength in concretes with no coarse aggregate replacement but with 50% mixed fine aggregate compared with the reference concrete. Corinaldesi et al. [19], Malesev et al. [20] and López et al. [21] noted that the elastic modulus decreased between 15% and 44.8% compared with the reference concrete, where the recycled coarse aggregate had a greater negative influence. Lovato et al. [22] even reveals that for the same level of axial compressive strength, higher than 20 MPa, and despite the greater consumption of cement, the costs are similar to those of the reference concrete and only vary by around 20%. Agrela et al. [23] suggest the following classification for RA: Recycled concrete aggregate, with >90% concrete; Mixed recycled aggregates (MRA), with between 30% and 10% ceramic content; and recycled ceramic aggregates, with >30% ceramic content. Regarding flexural strength [18], it is similar in conventional concrete and recycled concrete with less than 75% replacement. In higher percentages, the use of recycled materials in both fractions causes up to a 15% strength loss. Recycled concretes that have both coarse aggregate recycled concrete and recycled sand are suitable for their use in structural concretes with a characteristic strength of 30 MPa.

Regarding mixed recycled aggregates (MRA), due to the volume they represent of the total amount of C&DW (~67% of the total in Spain), in the past decade they have aroused the interest of the scientific community. There is currently a scientific–technical shortcoming in the feasibility of assessing the coarse and/or fine fractions in the design of concretes. In this regard, Sáez del Bosque et al. [24] revealed that the elastic modulus of the ITZ (Interfacial Transition Zone) varies with the type of materials that are present in the recycled aggregate, with ITZs associated with organic components (such as wood, plastic and asphalt) having a lower minimum elastic modulus value depending on the content of ceramic and concrete particles. Recently, Martínez-Lage et al. [25] recorded that the addition of 100% of MRA entailed saving more than 35% of waste generation and 50% of abiotic depletion. Martínez-Lage et al. [26] revealed that the decrease in compressive strength and the deformation modulus vary from 20% to 30% and 30% to 40%, respectively, in concretes with 100% coarse recycled aggregate replacement rate. Meanwhile, Poisson’s ratio, which is independent of the rate of replacement, varied from 0.14 to 0.2. Mas et al. [27] revealed that compressive strengths, 90 days later and with 30% coarse aggregate replacement, decreased by 23.8% for CEM II cements and 13.5% for CEM III/A. The strength increased, with 50% replacement, by +7.5% for CEM V/A cement (Series 1-CEM V). López et al. [28] produced non-structural concretes with 100% coarse aggregate replacement using concretes with low strength (15 MPa), using 200 kg cement/m^3^ of concrete. Meanwhile, Medina et al. [29] revealed that the use of MRA in concretes with a 25% coarse aggregate replacement rate has no effect on the absorption capacity. However, at a replacement rate of 50%, the sorptivity of recycled concretes is 10–20% higher than the reference concrete. Cantero et al. [30] concluded that in concretes with coarse MRA, the higher the replacement rate, the lower the density in a fresh state and the higher the air content: in concrete with a 100% rate of replacement, the density was 7% lower and the air content 37% higher than the reference concrete. It also backs its use in concretes, with a characteristic strength of 30 MPa.

As a result of the research conducted, recycled aggregates have been added unevenly in the regulation of concrete in different countries. Table 1 shows the type of recycled aggregate, fraction and maximum percentage (minimum and maximum value) allowed by each continent, as well as the type of concrete (structural or non-structural) that can be crafted using recycled aggregates. It also shows that the common aspect is that all continents allow the use of the coarse fraction of RCA and RMA, whose maximum limit of replacement ranges from 15% to 100% for structural or non-structural concretes. Regarding the fine fraction of RCA, it reveals that, with the exception of the American continent, its use is allowed in the eco-design of new concretes. In addition, fine MRA is only allowed for non-structural uses. In this context, it is worth noting that in South American countries [31], the technical regulation does not yet allow the use of recycled aggregate (RCA/MRA), due to the lack of trust it generates and the absence of authorised managers of C&DW who obtain optimal quality recycled aggregates that are durable. 

In this context, the main novelty of this research study lies in delving into the progress of the knowledge on the joint valorisation of the coarse and fine fraction of mixed recycled aggregate, as research conducted heretofore had focused mainly on the RCA fraction and, to a lesser extent, on the coarse fraction of MRA. There are no prior studies that simultaneously use fine and coarse MRA to produce structural recycled concretes. Likewise, this study contributes to the development of regulation or guides that make it possible to, depending on the quality of the fMRA (their composition and intrinsic properties), establish the applications where they could be used, such as the concrete industry, defining the minimum parameters it must meet to comply with the building code and the areas of environmental exposure where new concretes can be used, as well as other applications in the scope of the construction sector. As stated by Plaza et al. [18], the partial replacement of natural aggregate with coarse recycled aggregate alone or together with fine recycled aggregate (RCF) from concrete wastes has a generally beneficial effect on eco-efficiency (the relation between compression, tensile strength, bending and CO_2_ emitted), with values that are similar or greater than those shown by HP (conventional concrete). The use of coarse recycled aggregate concrete and RMF (recycled mixed fine aggregate) has a beneficial effect on eco-efficiency to split tensile strength (strength/CO_2_ emission ratio), whereas the eco-efficiency is slightly lower (<3%) than HP in terms of compressive and flexural strength. The benefits of using coarse and/or fine recycled aggregate to partially replace natural aggregate are not only a decrease in the emissions of CO_2_ when manufacturing concrete but also the significant mitigation of environmental issues triggered by gathering the necessary waste.

In this line of research, the main goal of this research was to further deepen scientific and technical knowledge on the simultaneous use of the fine and coarse fractions of the mixed fraction of construction and demolition waste as components of the granular skeleton of recycled concretes. Doing so required characterising the physical (density, entrained air and consistency) and mechanical (compressive, tensile and flexural strength) characterisation, as well as the watertightness under pressure, of the new eco-concretes, which include 0%, 25% and 50% of coarse MRA and/or 0%, 25% and 50% of fine MRA, which can be used for applications in the field of civil engineering and construction.

## 2. Materials and Methods

### 2.1. Materials

The natural aggregate (NA) used was crushed siliceous greywacke with sharp edges and layered shapes (see Figure 1) which comes in two granulometric fractions: (i) natural 0/6 mm sand (fNA); and (ii) natural 6–12 mm gravel (cNA). This aggregate meets all the requirements of European standard EN 12620 for aggregates used to manufacture concretes. Table 2 shows the physical and mechanical properties of the aggregates used in the study, as well as the requirements stipulated by European standard 12620 [32] and EHE-08 for aggregates used to manufacture concretes [33].

The MRA used came from the ARAPLASA C&DW management plant, located in the north of the province of Cáceres (Spain), which came, like natural aggregates, in two fractions: (i) recycled mixed 0/6 mm sand (fMRA) and (ii) mixed recycled 6/12 mm gravel (cMRA). Regarding their morphology, they are characterised by preferably having rounded and slightly layered shapes (see Figure 1). Regarding their composition (see Table 3), the gravel is characterised by being comprised by ~88% debris from unbound concrete, mortar and aggregate, as well as ~11% ceramic material (tile, blocks, bathroom fittings, etc.) and other minority components (<1.3%), mainly floating particles (plastic and wood), gypsum and glass, in a percentage that makes up less than 1% of the weight. In addition, it is worth noting that fMRA was obtained from the same MRA as cMRA, with the former having a reddish colour due to the presence of ceramic fines.

Figure 1 and Figure 2 shows the granulometric distribution of the natural and recycled aggregates, revealing that they all have a continuous grain size. Likewise, it reveals that regardless of their nature, sands are within the granulometric spectrum recommended by EHE-08 for manufacturing concretes, as is the amount of particles that pass through the 0.063 mm sieve, which is less than 10% of the weight, the maximum limit stipulated by EHE-08 for crushed sands of a siliceous nature.

The Portland cement used was a CEM I 42.5 R that met all the physical, chemical and mechanical requirements established by European standard EN 197-1 [38] and was supplied by the plant of the Lafarge Holcim group located at Villaluengo de la Sagra in the province of Toledo (Spain). Lastly, we used superplasticiser additive FUCHS BRYTEN NF, a water-reducing additive made of modified polycarboxylate with a base of water and a brownish colour. It is free of chloride, has a ~20% content of solids, a density of 1.1 g/cm^3^ and pH = 8.0 and was supplied by FUCHS Lubricantes S.A.U.

### 2.2. Concrete Properties Studied

Table 4 lists the properties analysed in fresh and hardened states, as well as the standard that describes the trial methodology and size of the samples used to assess the property being studied.

### 2.3. Mix Design

Nine mixes were manufactured: (i) Conventional concrete, with 100% NA (M1); (ii) Concrete with 25% fine MRA (M2); (iii) Concrete with 50% fine MRA (M3); (iv) Concrete with 25% coarse MRA (m4); (v) Concrete with 25% fine MRA and 25% coarse MRA (M5); (vi) Concrete with 50% fine MRA and 25% coarse MRA (M6); (vii) Concrete with 50% coarse MRA (M7); (viii) Concrete with 25% fine MRA and 50% coarse MRA (M8); (ix) Concrete with 50% fine MRA and 50% coarse MRA (M9).

In order to design and formulate the mixes we used the mix-design rules [46], taking as baseline data a design characteristic strength (*f_ck_*) of 25 MPa (C25/30), a maximum aggregate size of 12.5 mm, a constant effective water–cement ratio (w/c) of 0.45 and an S2 target workability as established in EN 206-1 [47], which is equivalent to a 70 ± 20 mm slump. Likewise, dry aggregates were taken into consideration in this dosage process, as well as the water absorption of the recycled aggregates in the first 10 min of being immersed in water (~70% of the total water absorption after 24 h). With this strategy, we guaranteed that all mixes had the same amount of water available to hydrate the cement, regardless of their granular skeleton. In addition, 5.89 kg/m^3^ of additive was added in order to achieve a suitable workability with this w/c ratio.

All the mixes designed met the dosing requirements (minimum cement content: 300 kg/m^3^ and maximum effective w/c ratio: 0.55) stipulated by European standard EN-206-1 [47] for durability class XC2. The mixes studied (Table 5) were prepared in an 85-litre laboratory vertical shaft mixer using the following procedure: (i) the coarse aggregate (cNA and cMRA) was mixed for 30 s; (ii) the fine aggregates (fNA and fMRA) were added and the materials were mixed for 30 s; (iii) the binder (cement) was added, mixing it for 60 s; (iv) then 80% of the mixing water was added as well as the superplasticiser additive, mixing it for 45 s and (v) the remaining water was added and mixed for 240 s. Lastly, the samples were manufactured and cured following European standard EN 12390-2 [48].

Figure 3 summarizes the manufacturing process and tests carried out on the studied mixtures.

## 3. Results and Discussion

### 3.1. Properties in a Fresh State

#### 3.1.1. Consistency

Table 6 shows the results of consistency, entrained air content and density of the concretes designed. Regarding consistency, all mixes were within the target design workability (50 ≤ S2 ≤ 90 mm), which reveals that adding the mixed recycled fine and/or coarse fraction does not have a negative effect on this property. This result is in line with Plaza et al. [18] who observed that the simultaneous addition of recycled concrete aggregate and concrete or mixed sand does not lead to a decline in the workability of recycled concretes. This behaviour is in line with Agrela et al. [23] and Medina et al. [29] who respectively suggested, as strategies to mitigate the negative effect that the higher water absorption of recycled aggregates has on this property, to pre-saturate them before the mixing process or to add the water initially absorbed by these recycled aggregates to the dosage.

Figure 4 shows the concrete slump test of conventional concrete (M1) and concrete with a higher content of MRA, highlighting that the individual or simultaneous use of the fractions does not have a negative effect on this property.

#### 3.1.2. Entrained Air

Table 6 lists the amount of entrained air in fresh concrete, revealing that the addition of fine and/or coarse MRA caused an increase in this property, with the value for M3 (50% fMRA), M7 (50% cMRA), M8 (25% fMRA + 50% cMRA) and M9 (50% fMRAf 50% cMRA) being 1.9, 1.4, 2.1 and 2.79 times higher than that registered for mix M1 (100% NA), respectively. This performance could be connected with [49,50] (i) higher water absorption by the MRA, (ii) lower density of the adhered mortar present in the recycled aggregate due to the presence of air bubbles within, (iii) the rougher texture of MRA compared with NA and (iv) the presence of microcracks inside the MRA which are not connected to the aggregate’s permeable pores. These results are in line with the observations by Cantero et al. [30] and Plaza et al. [18], who registered an increase of this property when adding contents of up to 100% MRA and 100% RCA, respectively.

Figure 5 reveals the linear relationship that exists between this property and the percentage of recycled sand for different percentages of coarse aggregate replacement (0%, 25% and 50%), with all cases having an *R*^2^ ≥ 0.99. This trend was previously registered by Yaprak et al. [49], who added from 0% to 100% of recycled concrete sand.

#### 3.1.3. Density

Table 6 lists the density values for fresh concrete. It reveals a decrease of said property in connection with the percentage of addition of mixed recycled aggregate (fMRA and/or cMRA). These decreases reached their maximum value in mixes that had both fractions added at 50% simultaneously; as for mixes M8 and M9, these values were 5.4 and 5.3 compared with mix M1, respectively. This was also the case with the mix that added 25% cMRA + 50% fMRA (M6), with a 5.8% decrease compared with M1. This decrease is in line with Brito et al. [50], who registered decreases of up to 10% for concretes that had up to 100% of MRA. This behaviour would be connected with the lower density of recycled aggregates due to the presence of adhered mortar and ceramic material, as well as the apparent higher water/cement ratio of the new concrete mixes designed [49,51].

Regarding the values obtained, it is worth noting that they were within the 2430–2300 kg/m^3^ and 2430–2220 kg/m^3^ ranges of values for concretes that have different percentages of recycled concrete aggregate [52,53] and mixed recycled aggregate [51], respectively. Lastly, this observed trend is in line with research previously registered by other authors: Plaza et al. [18] registered densities of 2428.56 kg/m^3^ for conventional concretes, 2367.38 kg/m^3^ for concretes with a 100% coarse aggregate replacement rate and 2310.51 kg/m^3^ for mixes with 100% recycled coarse aggregate replacement and 50% fMRA; César Medina et al. [52] registered a density of 2347 kg/m^3^ with 25% of cMRA replacement and densities of 2335 kg/m^3^ with 50% cMRA replacement. A. González et al. [51], with high-performance recycled concrete aggregates with 20% and 50% cMRA replacement, obtained densities of 2430 kg/m^3^ and 2340 kg/m^3^, respectively.

### 3.2. Properties in a Hardened State

#### 3.2.1. Density

Table 7 lists the values of apparent density in hardened concretes analysed following 28 days of curing, once again revealing a decrease of this property with the percentage of addition of MRA (fMRA and/or cMRA), due to the lower density of this new typology of recycled aggregates compared with natural aggregates (Table 2). This decrease was between 1.4% and 5.7% compared with the reference concrete (M1). In this context, it is worth noting that said decrease is in line with the 3.3–5.0% range registered by other authors who added up to 100% of MRA [49]. Regarding the values obtained, it is worth stressing that they are within the range of values (2450–2270 kg/m^3^) registered previously by other authors who added mixed recycled aggregates [10,29,30].

#### 3.2.2. Compressive Strength

Table 7 shows the evolution of the compressive strength of the various mixes analysed after 7, 28 and 90 days. It shows that regardless of the type of concrete, (i) compressive strength increases with the curing time, an evolution which is very similar to that displayed by the reference concrete, (ii) the relative compressive strength after seven days is 75.7–83.6% of that obtained after 28 days, a similar percentage to the figure obtained (70%) in Portland cement concretes with no additions and a w/c ratio <0.45 [54] and to the 65–93% range of values registered by Bravo et al. [16] for concretes that partially added C&DW, and (iii) the average strength after 28 days is higher than the design strength of 25 MPa. Therefore, these new recycled concretes could be used as concretes for structural use.

Figure 6 shows the compressive strength variation of recycled concretes compared with conventional concrete for the various curing times. It reveals that for short times (t < 28 days), the addition of fMRA and/or cMRA caused a maximum performance loss of 10.7% compared with M1 and was suffered by the mix that had 50% fMRA + 50% cMRA (M9). However, at t = 90 days there was an increase in strength. The maximum increase was +19% over M1, observed in the mix that had 25% fMRA (M2). Likewise, concretes with 50% cMRA and 0%, 25% and 50% fMRA (M7–M9) had a better behaviour, recording increases ranging from 8.4% to 12.4% compared with M1.

These results reveal that the addition of cMRA individually (M4–M7) does not have a negative effect on this attribute, registering a small decrease (Δ_M4_ = −3.8%/Δ_M7_ = −5.6%) and a slight increase (Δ_M4_ = +3.5%/Δ_M7_ = 8.4%) compared with M1 after 28 and 90 days, respectively. This performance is in line with the observations of Cantero et al. [30], Poon et al. [54], Gomes et al. [55] and Lotfy et al. [56], who revealed that for recycled coarse aggregate replacement percentages ≤ 50%, there are no significant differences with conventional concretes.

Regarding the individual addition of fMRA (M2 and M4), we observed that the addition of 25% in weight (M2) led to a slight increase in compressive strength, reaching +3.8% and +19.1% compared with M1 after 28 and 90 days, respectively. This behaviour is in line with the study of Bravo et al. [10], who established that type I recycled fine aggregates (Rc + Ru ≥ 80% and other components ≤20%) caused variations in compressive strength ranging from +3.8 to −12.5 after 28 days and in concretes with a replacement percentage of 25%. Likewise, this result was also obtained by other authors [57,58,59,60], who registered 3.7%, 5% and 16% increases compared with conventional concrete for 10%, 30% and 50% additions of RCA, respectively. This increase was connected with the filler effect of fine RCA and their rough texture and shape, which allowed for a better packing of the granular skeleton [61,62].

Regarding the 50% fMRA replacement (M3), the study showed that after 28 days the resistance remained constant, whereas after 90 days, there was a slight increase of +8.1% compared with M1. This behaviour is better than that established by Bravo et al. [63] for a 50% replacement with type-I sand comprised by Rc + Ru = 83.7%, Rb = 0.9% and Rg = 15.4. These authors registered an attribute loss ranging from −6.1% to −13.3% compared with conventional concrete, for a 50% percentage of replacement. This improved behaviour of M3 is linked to the better quality of fMRA (Table 2) compared with the type-I sand studied by said authors.

Likewise, Figure 5 reveals that mixes M5, M6, M8 and M9, which simultaneously incorporated cMRA (25% and 50%) and fMRA (25% and 50%) in their composition, had a worse performance after 28 days, registering a 5.5%, 5.0%, 8.0% and 1.4% decrease compared with M1, respectively. These decreases are lower than the ~7% and ~17% observed by Plaza et al. [18], who added 50% coarse RCA + 50% fine RCA and 50% coarse RCA + 50% mixed sand, respectively. In addition, it is worth noting that similar to when MRA were added individually, an improved performance was registered for the concretes after 90 days of curing, with mix M9 (cMRA = 50% + fMRA = 50%) reaching a maximum increase of 12.4% compared with M1.

This improved behaviour of the new concretes (M2–M9) at curing times greater than 28 days could be connected with the pozzolanic activity of the fine fraction (sizes < 0.063 mm) present in fMRA, where there are mainly ceramic fines. This aspect was previously registered by Medina et al. [60,61] and Asensio et al. [62,63], who analysed the pozzolanic performance of the fine fraction of C&DW with variable compositions (26.5% ≤ SiO_2_ ≤ 70.5%, 4.4% ≤ CaO ≤ 24.5%, 5.8% ≤ Al_2_O_3_ ≤ 18.5%), observing that they have a lime fixation capacity after 28 days of 53.3–82.1%. These values are lower than silica fume (~90%) and higher than fly ash (~45%).

In addition, this behaviour could also be linked to (i) the presence of anhydride cement particles [64,65,66] in the mortar adhered to the fMRA that will become hydrated, generating calcium silicate hydrate (C-S-H) which will positively contribute to the mechanical behaviour and (ii) the attributes of existing ITZs among the Rb components of MRA (cMRA and fMRA)/paste that can have a thickness equal to or lower than the one that natural aggregates normally have (e_ITZ_ = 10–50 μm), as shown previously in the studies by Medina et al. [29] and Sáez del Bosque et al. [24] in concretes that had coarse fractions of ceramic aggregate from bathroom fittings and coarse mixed recycled aggregate, respectively.

Figure 7 shows the appearance of concretes M1, M3, M6 and M9 after being subjected to the compression test after 28 days, revealing that the type of failure was similar in all of them and that their morphology can be classified as suitable according to European standard EN 12390-3 [65,67].

Lastly, Figure 6 verifies that the aggregate/paste ITZ is the area where the cracks preferentially begin and then spread when a concrete item is subjected to an external force that surpasses its operational status limit [68]. This happens because this area is the weakest and the one where the most stress concentrates, as observed by Medina et al. [67] by simulating the stress in the coarse/paste aggregate ITZ subjected to compressive forces [69].

#### 3.2.3. Splitting Tensile Strength

Table 8 shows the tensile strength of concretes tested after 28 days and the variations in strength compared with conventional concrete (M1) and compared with the concretes that exclusively had cMRA (M4 and M7). It reveals that the mixes that only had fMRA (M2 and M3) experienced a slight increase in tensile strength between 8.7% and 5.5% compared with M1, respectively. This behaviour is in line with the observations of Ahmed et al. [68] and Kirthika et al. [69], who revealed that the use of fine MRA in percentages lower than 50% did not lead to a significant loss of performance (<10%) in the new concretes.

Regarding the mixes whose composition included 25% cMRA (M4) and 50% cMRA (M7), it is worth noting that they registered a small decrease of 7.0% and 4.6% compared with M1, respectively. These losses are similar to those registered by Cantero et al. [30] in concretes with 25% and 50% of mixed recycled aggregate and lower than the 12% and 14% losses registered previously by other authors who [70,71,72,73,74,75] analysed concretes with 100% of recycled aggregates, which had 10% ≤ Rb ≤ 14%.

Regarding the mixes that include both fractions simultaneously, we observed that the addition of fMRA had a positive effect by (i) lessening the decrease in strength of the mixes (M5, M6, M8 and M9) compared with the mixes that only included cMRA (M4 and M7), with the decrease reaching 1.45% ≤ Δfcmt ≤ 4.06%, and (ii) increasing their strength compared with mixes that only had cMRA (M4 and M7), with this change ultimately being 3.12% ≤ Δfcmt ≤ 6.23%. This behaviour is better than that observed previously by Plaza et al., who registered a 7.3% and 11.0% decrease compared with conventional concrete in concretes with 25% coarse RCA + 50% fine MRA and 50% coarse RCA + 50% fine MRA, respectively. This same tendency was revealed for concretes with 50% coarse RCA + 50% fine RCA, again registering decreases ~11% compared with conventional concrete [76].

Lastly, it is worth noting that the failure mechanism was the same in all tested concretes, causing a brittle failure that led to the tested mixes splitting in half. In addition, we observed that (i) the failure surface obtained is irregular, confirming the existence of intact coarse aggregates (cNA or cMRA) and thus again revealing that the failure emerges from the ITZ of the coarse and/or fine aggregates and the cement paste, and (ii) the granular skeleton (fine and coarse aggregates) is distributed homogeneously, regardless of whether the aggregate is natural or recycled (Figure 8).

#### 3.2.4. Flexural Resistance

Table 8 lists the flexural strength values of the tested concretes after 28 days, and the variations in strength compared both with conventional concrete (M1) and with the concretes that exclusively had cMRA (M4 and M7). It reveals that the mixes that only had fMRA (M2 and M3) experienced a slight increase in flexural strength of +8.9 and +8.6% compared with M1, respectively. This result is in line with the prior observations of Ahmed [76] and Kirthika [77], who registered 6.7% and 3.2% increases compared with conventional concrete for a fine MRA replacement percentage of 50%, respectively.

Regarding mixes with 25% cMRA (M4) and 50% cMRA (M7), it is worth noting that they displayed an uneven behaviour, with mix M4 registering a 7.3% increase compared with M1 and M7 showing a 3.4% decrease compared with M1. This behaviour was similar to that observed by Cantero et al. [30], who established that for replacement percentages lower than or equal to 50% of coarse MRA, there were no significant variations of this mechanical attribute (Δfcmf ≤ 10% compared with conventional concrete).

Regarding the mixes that simultaneously incorporate both fractions, we observed that the addition of fMRA had a positive effect by improving the behaviour of mixes with cMRA (M4 and M7). It is worth noting that mixes M5 and M6 registered a 4.4% and 2.7% increase compared with M4, and mixes M8 and M9 showed a 3.8% and 10.0% increase compared with M7. Likewise, all of them had a similar or improved behaviour compared with that observed in conventional concrete with 100% natural aggregate (cNA and fNA).

Lastly, all samples tested, regardless of the composition of their granular skeleton, had failures due to the formation of a crack in the middle part of the span, rising from the part being pulled (the lower part of the sample) to the part being compressed [77] (the highest part where the load is applied). Likewise, observing the cracked area, we once again saw that the failure took place along the aggregate/paste transition area, with the aggregates being detached from the matrix.

#### 3.2.5. Water Penetration Depth under Pressure

Figure 9 shows the results obtained regarding the maximum and mean depth of the mixes analysed.

Regarding the individual incorporation of fMRA (M2 and M3), we observed that these mixes experienced a larger decrease both in maximum (Pmax) and mean (Pmed) water penetration, with the decrease ending up being between 16.1% ≤ Pmax ≤ 44.6% and 47.7% ≤ Pmed ≤ 52.9% compared with conventional concrete (M1). This greater watertightness could be connected with the pozzolanic activity of ceramic fines (<0.063 mm) of the fMRA, as well as with the hydration of anhydrous cement [78] present in the fines from fMRA mortars that give them a certain hydraulic activity and thus lead to a more sealed and tortuous pore structure.

Regarding the use of cMRA, we observed that an addition of 25% (M4) and 50% (M7) caused an uneven behaviour in these two properties. In the case of M4, the table shows that Pmax and Pmed experienced a slight (2.8%) and small (20.1%) decrease compared with M1, respectively. This behaviour is in line with the prior observations of Mas et al. [77], who observed that the depth remained constant for coarse MRA replacement percentages ≤25%. Regarding mix M7, it experienced an increase of Pmax (2.4%) and a slight decrease of Pmed (6.4%). This behaviour could be connected with the fact that the microcracks present in MRA have a greater impact on Pmax than Pmed.

Regarding the simultaneous use of cMRA and fMRA as the granular skeleton of the concretes, the table shows that only the 25% cMRA + 25% fMRA mix (M5) made it possible to obtain an increase in water penetration, causing a 35.8% and 33.9% decrease in Pmax and Pmed compared with M1, respectively. This result, as happened with the other properties studied, makes it possible to say that there is no performance loss for MRA replacement percentages lower than or equal to 25%.

In this context, it is worth noting that for weight percentages of cMRA and fMRA greater than 25%, there was an increase in water penetration, which could be explained by the beneficial effect of its rougher surface texture enabling a better entry (similar ITZs) in the cement matrix, as well as the pozzolanic activity of the <0.063 mm fraction not being able to compensate for the negative effect that its intrinsic properties have on water penetration (lower density, higher water absorption and the presence of microcracks in its microstructure).

The values obtained are under the limits established in chapter VII “Durability” Section 37.3.3 “Resistance to water penetration” of the Spanish Code on Structural Concrete (EHE-08), which establishes that a concrete is watertight enough for a given type of exposure (IIIa, IIIb, IIIc, IV, Qa, E, H, F, Qb, Qc) when its maximum and mean penetration depth is lower than 50 or 30 mm and 30 or 20 mm, respectively, depending on the type of environment (see Figure 9). This highlights that recycled concretes have a porous structure that guarantees watertightness and a suitable durability throughout their useful life against this transport mechanism.

Lastly, Figure 10 shows the penetration front of the samples manufactured for concretes M1, M3, M6 and M9. As can be seen, the water penetration outlines have a similar morphology, and there are no visible differences between the mixes manufactured with NA and cMRA and/or fMRA.

#### 3.2.6. Analysing the Concrete Manufacturing Costs

Table 9 lists the economic study of the cost of manufacturing the concretes analysed in order to reveal the financial aspects of this study. In this context, it is worth noting that the price of natural aggregates in Spain is lower than in other countries, as this country is characterised by having a high availability of natural resources, which enables the extraction of natural aggregates. These prices (EUR/t) will be much higher in countries with greater legal restrictions on extracting natural resources or with less availability, which would facilitate the recovery of recycled aggregates in the concrete industry from an economic point of view. This table shows that mix M9, with 50% fMRA + 50% cMRA, is the cheapest, leading to a −8.03% decrease compared with natural aggregate (M1). This result is in line with the prior observations of other authors, who registered decreases of under −50% for recycled coarse aggregate replacement percentages between 50% and 100% [78].

Lastly, in addition to these economic savings, one must consider the positive environmental effect, especially in terms of kgCO_2_eq/kg, that the correct management of MRA and a decrease in the extraction of natural aggregates entail.

## 4. Conclusions

The conclusions drawn from this research study are:-Coarse (cMRA) and fine (fMRA) mixed recycled aggregate meet the physical and mechanical requirements included in the relevant regulation on aggregates for manufacturing concretes.-The workability of recycled concretes (M2–M9) is not impacted by the addition of mixed recycled aggregate, regardless of its replacement percentage, with all mixes showing an S2 consistency (50–90 mm).-The addition of mixed recycled aggregate (cMRA and/or fMRA) causes a linear increase in the entrained air content in a fresh state of 1.9, 2.8 and 2.8 times the M1 mix in mixes M3 (50% fMRA), M6 (25% cMRA + 50% fMRA) and M9 (50% cMRA + 50% fMRA), respectively.-The density of recycled concretes is lower than that of conventional concrete, with the loss of density increasing as the recycled aggregate replacement percentage rises and with greater intensity when using mixed aggregate. This behaviour is similar both in a fresh and hardened state.-The compressive strength of recycled concretes is lower than that of conventional concrete, with M9 (50% cMRA + 50% fMRA) having the greatest drop (~11%) compared with M1 after 28 days of curing. After 90 days, recycled concretes have a better behaviour than conventional concretes. The addition of fMRA has a positive effect, establishing that the optimal replacement percentage is 25% in weight, regardless of the percentage of cMRA.-The compressive strength of all concretes is higher than the design strength (*f_ck_* = 25 MPa).-The indirect tensile strength experiences a slight increase with the addition of fMRA, with the maximum increase (~9%) compared with M1 corresponding to the concrete that incorporates 25% of fMRA (M2). The addition of cMRA causes a slight loss (~7%) compared with conventional concrete (M1). This loss is softened with the simultaneous addition of fMRA, obtaining a −0.3% decrease compared with M1 for mix M9 (50% cMRA + 50% fMRA).-The flexural strength of the new concretes is greater than conventional concrete, with the highest increases (6.3–12.0% compared with M1) belonging to mixes M2, M5 and M9. The addition of 50% of cMRA (M7) causes a −3.4% decrease compared with M1.-All the concretes designed have a watertight structure under pressure, meeting the maximum and mean depth requirements of the relevant regulation.-The (maximum and mean) water penetration under pressure decreases slightly for mixes M2, M3, M4 and M5, whereas an increase was registered in one (M6, M7 and M8) or both depths (M9) for all remaining concretes.-The optimal individual or simultaneous replacement percentage of fMRA and cMRA is 25% in weight, in light of the results obtained in this research study.-These results reveal the need for future research that addresses the behaviour of these concretes from the viewpoint of their durability properties, as well as investigating different types of mixed recycled aggregates with which to manufacture concretes.-Lastly, this research will positively contribute to the addition of these mixed recycled aggregates to the concrete-related stipulations of structural codes.

## Figures and Tables

**Figure 1 materials-14-04762-f001:**
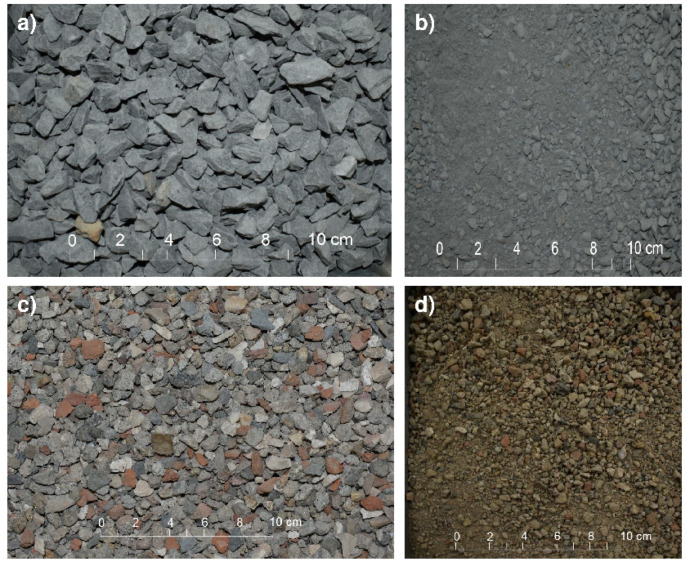
View of the aggregates used to manufacture concretes. Legend: (**a**) View of natural aggregate; (**b**) View of natural aggregate after grinding; (**c**) View of recycled aggregates and (**d**) View of recycled aggregates after grinding.

**Figure 2 materials-14-04762-f002:**
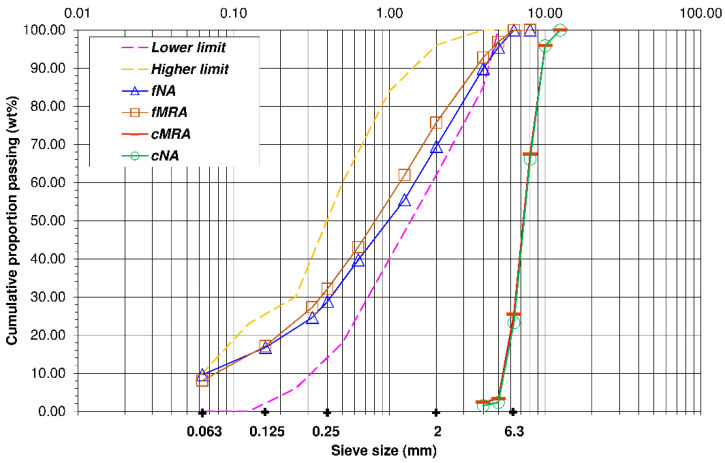
Granulometric distribution of aggregates (NA and MRA).

**Figure 3 materials-14-04762-f003:**
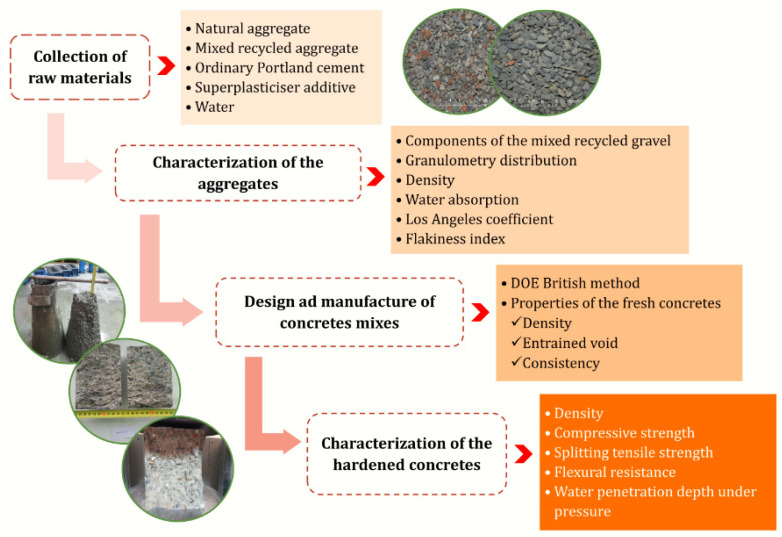
Flow diagram corresponding to the manufacturing process and tests carried out on the studied mixtures.

**Figure 4 materials-14-04762-f004:**
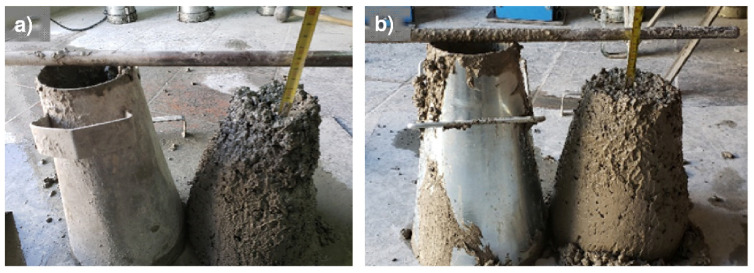
Concrete slump test: (**a**) Concrete with 100% NA (M1) and (**b**) Concrete with 50% fMRA and cMRA (M9).

**Figure 5 materials-14-04762-f005:**
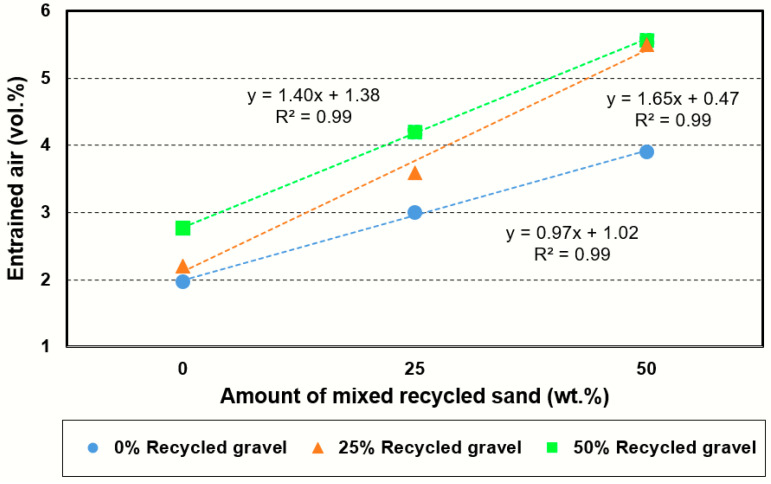
Connection between the percentage of mixed recycled sand and entrained air.

**Figure 6 materials-14-04762-f006:**
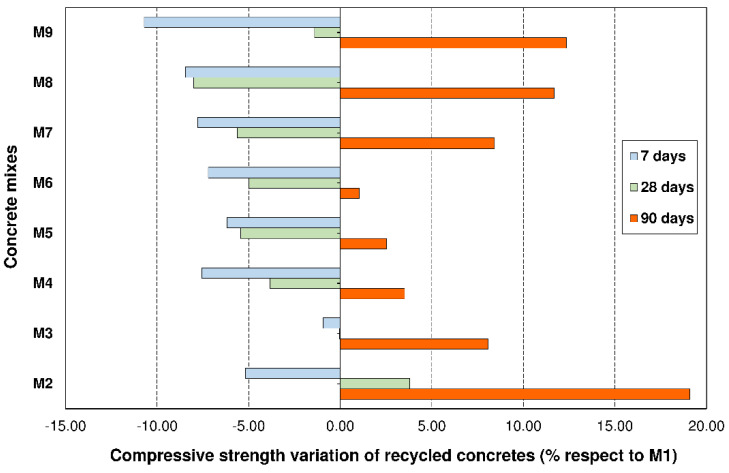
Compressive strength variation of recycled concretes (M2–M9) compared with conventional concrete (M1).

**Figure 7 materials-14-04762-f007:**
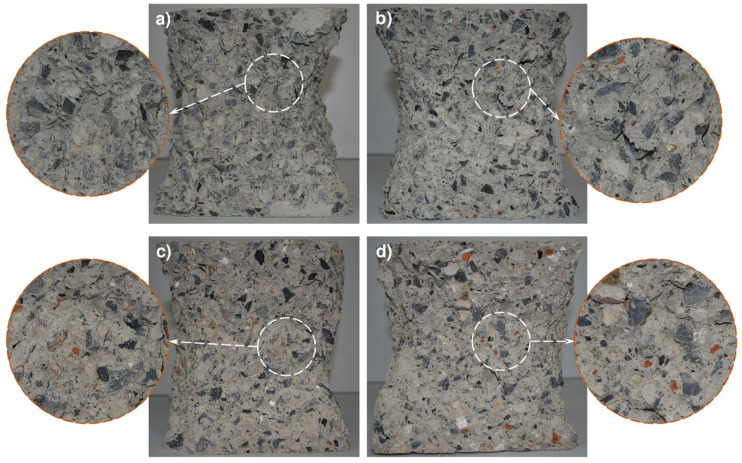
Type of failure of the concretes: (**a**) M1; (**b**) M3; (**c**) M6; and (**d**) M9.

**Figure 8 materials-14-04762-f008:**
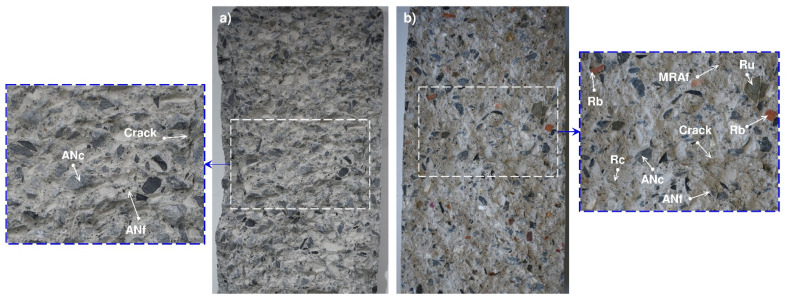
Appearance of the concretes when performing the tensile test 28 days after producing the mortar: (**a**) conventional concrete (M1) and (**b**) concrete with 50% cMRA + 50% fMRA (M9).

**Figure 9 materials-14-04762-f009:**
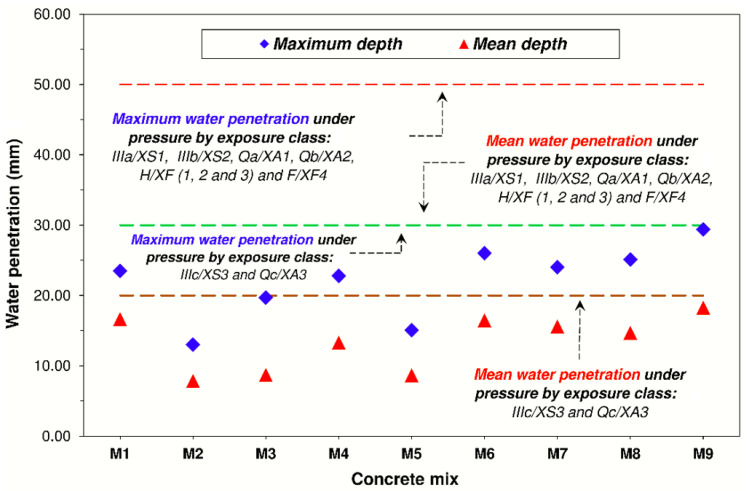
Water penetration of the concretes under pressure. Limits established in EHE-08 and EC-2 (Note. IIIa: marine class—subclass: aerial; IIIb: marine class—subclass: submerged; IIIc: marine class—subclass: tidal and splash zones; Qa: aggressive chemical class—subclass: weak; Qb: aggressive chemical class—subclass: average; Qc: aggressive chemical class—subclass: strong; H: with frost class—subclass: without deicing salts; and F: with frost class—subclass: without deicing salts).

**Figure 10 materials-14-04762-f010:**
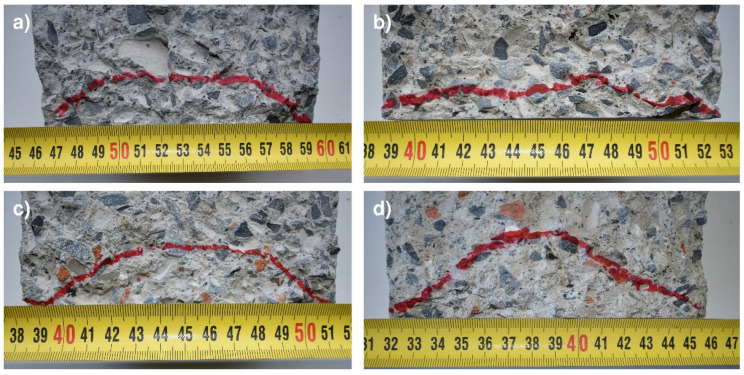
Penetration fronts of the concretes: (**a**) M1; (**b**) M3; (**c**) M6; and (**d**) M9.

**Table 1 materials-14-04762-t001:** Recycled aggregates in international regulation. Typology and maximum rate of incorporation.

Continent	Material	Fraction	% Allowed		Type of Concrete
			Minimum Value	Maximum Value	
Asia (China, Korea and Japan)	RCA	Coarse	20	100	Structural
		Fine	30	100	Structural
		Coarse Fine	0 0	100 30	Non-structura Non-structural
Australia	RCA	Coarse	-	30	Structural
	MRA	Coarse	-	100	Structural
Europe (Belgium, Germany, Italy, Denmark, Holland, Portugal, Switzerland, United Kingdom, France, Spain)	RCA	Coarse	15	100	Structural
		Coarse/Fine	-	100	Structural
	MRA	Coarse	25	100	Structural
		Fine	-	20	Non-structural
America (Brazil)	RCA	Coarse/Fine	-	100	Non-structural
	MRA	Coarse/Fine	-	100	Non-structural

Note. RCA: recycled concrete aggregate; MRA: mixed recycled aggregate.

**Table 2 materials-14-04762-t002:** Physical and mechanical properties of the aggregates.

Property	Aggregates				EN-12620/ EHE08
	fNA	cNA	fMRA	cMRA	
Dssd (kg/m^3^) [34]	2.76	2.74	2.70	2.42	-
WA_24_ (wt %) [34]	1.18	0.88	5.39	6.28	<5
LC (wt %) [35]	-	16	-	32	<40
FI (wt %) [36]	-	21		10	<35

Note. fNA: natural sand; cNA: natural gravel; fMRA: mixed recycled sand; cMRA: mixed recycled gravel; Dssd: dry saturated surface density; WA_24_: water absorption coefficient after 24 h; LC: Los Angeles coefficient; and FI: flakiness index.

**Table 3 materials-14-04762-t003:** Components of the mixed recycled gravel (cMRA) (EN 933-11 classification [37]).

Class	Type	Content (% Weight)
Rc	Concrete and mortar	43.98
Ru	Natural stone	43.94
Rc + Ru		87.82
Rb	Baked clay material	10.93
Ra	Asphalt	0.87
FL	Floating particles	0.02
G	Gypsum	0.34
X + Rg	Others and glass	0.02

**Table 4 materials-14-04762-t004:** The concrete properties analysed in fresh and hardened states.

Properties	Trial	Standard	Sample Size (mm)	Trial Duration (days)
Physical	Density	EN 12350-6 [39]	Cubic 150 × 150 × 150	Beginning
	Entrained air	EN 12350-7 [40]	-	Beginning
	Consistency	EN 12350-2 [41]	-	Beginning
Mechanical	Compression	EN 12390-7 [42]	Cubic 150 × 150 × 150	7, 28 and 90
	Traction	EN 12390-6 [43]	Cylindrical 100 ϕ × 200	28
	Bending	EN 12390-5 [44]	Prismatic 100 × 100 × 400	28
Durable	Penetration under pressure	EN 12390-8 [45]	Cylindrical 150 ϕ × 300	28

**Table 5 materials-14-04762-t005:** Composition of the concretes designed.

Materials (kg/m^3^)	Mix								
	M1	M2	M3	M4	M5	M6	M7	M8	M9
fNA	916.8	684.0	446.4	902.4	666.0	434.4	888.0	648.0	429.6
fMRA	0.0	228.0	446.4	0.0	222.0	434.4	0.0	216.0	429.6
cNA	993.2	988.0	967.2	733.2	721.50	705.90	481.0	468.0	465.4
cMRA	0.0	0.0	0.0	244.4	240.5	235.3	481.0	468.0	465.4
Cement	380.0	380.0	380.0	380.0	380.0	380.0	380.0	380.0	380.0
Water	224.4	228.9	231.1	232.8	237.4	241.5	243.9	247.5	252.3
Additive	5.9	5.9	5.9	5.9	5.9	5.9	5.9	5.9	5.9

**Table 6 materials-14-04762-t006:** Concrete properties in a fresh state.

Mix	Consistency (cm)	Entrained Air (vol %)	Density (kg/m^3^)
M1	6.00	1.97	2416.37
M2	6.00	3.00	2380.97
M3	5.17	3.90	2354.57
M4	5.50	2.20	2372.79
M5	5.50	3.60	2322.27
M6	6.50	5.50	2275.41
M7	6.50	2.77	2331.01
M8	5.67	4.20	2286.24
M9	5.17	5.57	2288.47

**Table 7 materials-14-04762-t007:** Density and compressive strength evolution.

Mix	D_28d_ (kg/m^3^)	a_cs7d_	σ	a_cs28d_	σ	a_cs90d_	σ
M1	2412.15	33.45	0.26	40.02	0.22	46.67	0.67
M2	2377.78	31.72	0.38	41.54	1.10	55.58	0.75
M3	2330.17	33.14	0.61	40.00	0.46	50.43	0.72
M4	2368.30	30.92	0.69	38.48	0.33	48.31	0.31
M5	2321.38	31.38	1.04	37.84	0.34	47.85	0.23
M6	2274.07	31.04	0.28	38.02	0.41	47.16	0.94
M7	2319.11	30.85	0.80	37.77	0.49	50.60	0.87
M8	2283.52	30.62	0.90	36.82	0.91	52.12	0.72
M9	2278.12	29.87	0.53	39.46	0.79	52.44	0.83

Note. D_28d_: density in a hardened state after 28 days; σ: standard deviation; a_cs7d_: average compressive strength after seven days; a_cs28d_: average compressive strength after 28 days; a_cs90d_: average compressive strength after 90 days. a_cs_ average compressive strength with a cubic sample, which must be multiplied by 0.90 to turn it into a cylindrical sample of 150 ϕ × 300 mm.

**Table 8 materials-14-04762-t008:** Tensile and flexural strength of concretes 28 days later.

Mix	fcmt (MPa)	Δfcmt (%) ^♣^	Δfcmt (%)	fcmf (MPa)	Δfcmf (%) ^♣^	Δfcmf (%)
M1	3.45 ± 0.09	-	-	3.82 ± 0.12	-	-
M2	3.75 ± 0.07	+8.70	-	4.16 ± 0.11	+8.90	-
M3	3.54 ± 0.13	+5.51	-	4.15 ± 0.23	+8.64	-
M4	3.21 ± 0.05	−6.96	-	4.10 ± 0.02	+7.33	-
M5	3.41 ± 0.06	−1.16	+6.23 **^♠^**	4.28 ± 0.09	+12.04	+4.39 **^♠^**
M6	3.31 ± 0.07	−4.06	+3.12 **^♠^**	4.21 ± 0.06	+10.21	+2.68 **^♠^**
M7	3.29 ± 0.06	−4.64	-	3.69 ± 0.27	−3.40	-
M8	3.40 ± 0.05	−1.45	+3.34 *****	3.83 ± 0.21	+0.26	+3.79 *****
M9	3.44 ± 0.06	−0.29	+4.56 *****	4.06 ± 0.20	+6.28	+10.03 *****

Note. fcmt: mean splitting tensile strength; fcmf: mean flexural strength; ^♣^ strength variation compared with M1; ^♠^ strength variation compared with M4; and * strength variation compared with M7.

**Table 9 materials-14-04762-t009:** Manufacturing cost of the mixes studied.

Component	Unit Price (EUR/t)	Concrete Mix								
		M1	M2	M3	M4	M5	M6	M7	M8	M9
fNA	6.79	6.23	4.64	3.03	6.13	4.52	2.95	6.03	4.40	2.92
fMRA	3.60	0.00	0.82	1.61	0.00	0.80	1.56	0.00	0.78	1.55
Can	6.54	6.50	6.46	6.33	4.80	4.72	4.62	3.15	3.06	3.04
cMRA	3.15	0.00	0.00	0.00	0.77	0.76	0.74	1.52	1.47	1.47
Cement	88.60	33.67	33.67	33.67	33.67	33.67	33.67	33.67	33.67	33.67
Water	0.50	0.11	0.11	0.12	0.12	0.12	0.12	0.12	0.12	0.13
Admixture	1.56	0.01	0.01	0.01	0.01	0.01	0.01	0.01	0.01	0.01
EUR/m^3^ concrete	-	46.51	45.72	44.76	45.49	44.59	43.67	44.49	43.51	42.78

## Data Availability

Data sharing not applicable.

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
