# Peer review of "The Design and Development of Recycled Concretes in a Circular Economy Using Mixed Construction and Demolition Waste"

_materials, 2021, doi:10.3390/ma14164762_

Round 1

Reviewer 1 Report

Minor revision
This manuscript introduces the research study which analyses the effect of adding fine – fMRA (0.25% and 50%) – and coarse – cMRA (0%, 25% and 50%) – mixed recycled aggregate both individually and simultaneously to the development of sustainable recycled concretes that require a lower consumption of natural resources In summary, the research is interesting and provides valuable results, but the current document has several weaknesses that must be strengthened in order to obtain a documentary result that is equal to the value of the publication.
General considerations:
(1)The document contains a total of 44 employed references, of which 32 are publications produced in the last 5 years (41%), 27in the last 5-10 years (34%), 19 than 5 years old (25%) , implying a total percentage of 75 % recent references. In this way, the total number is sufficient, and their actuality is high.
(2)The cost of this new technology and the value of the value have no detailed comparison, and readers cannot distinguish whether this technology has reasonable cost performance.
Title, Abstract and Keywords:
(3)The abstract is complete and well-structured and explains the contents of the document very well. Nonetheless, the part relating to the results could provide numerical indicators obtained in the research.
Chapter 1: Introduction
(4)The novelty of the study is not apparent enough. In the introduction section, please highlight the contribution of your work by placing it in context with the work that has done previously in the same domain.
(5)The first paragraph introducing the research topic gives a too simple, and even incomplete, view of the problems related to your topic and should be revised and completed with citations to authority references (Compressive properties of rubber-modified recycled aggregate concrete subjected to elevated temperatures; Experimental and Theoretical Investigation on the Thermo-Mechanical Properties of Recycled Aggregate Concrete Containing Recycled Rubber;Axial compression behavior of recycled-aggregate-concrete-filled GFRP–steel composite tube columns). 
(6)On a general level, the study of the proposed detection techniques is reasonable, and the explanation of the objectives of the work may be valid. However, the limitations of your work are not rigorously assumed and justified. 
Chapter 2: Materials and Methods
(7)A large number of English long sentences are used in this paragraph, it is recommended to break down into a few English simplicity, making readers better understand.
(8)The curve overlapping portion of Figure 2 is too much, and it is recommended to change a clear graph or a magnificent graph to facilitate the reader.

Chapter 3:    Properties in a fresh state
(9)In this chapter, a large number of professional nouns are mentioned, and the authors are recommended for appropriate explanations for some professional vocabulary so that the readers can better understand.ake the reader better understand.

Author Response

Revisor 1:

En primer lugar, queremos agradecer a los revisores sus comentarios. Los cambios realizados se resaltan en rojo.

(1) El documento contiene un total de 44 referencias empleadas, de las cuales 32 son publicaciones realizadas en los últimos 5 años (41%), 27 en los últimos 5-10 años (34%), 19 de 5 años (25%) , lo que implica un porcentaje total del 75% de referencias recientes. De esta forma, el número total es suficiente y su actualidad es alta.
(2) El costo de esta nueva tecnología y el valor del valor no tienen una comparación detallada, y los lectores no pueden distinguir si esta tecnología tiene un rendimiento de costos razonable.
Título, resumen y palabras clave:

Respuesta: Recuperar el tipo de áridos mixtos reciclados utilizados en este estudio, así como la fabricación y diseño de hormigones reciclados, no requiere la implementación de nueva tecnología en plantas de fabricación de hormigón. Estas plantas solo tendrían que ajustar los tiempos de mezcla y, en el proceso de diseño de las mezclas, añadir el agua que absorben inicialmente los áridos reciclados para preservar las propiedades reológicas.

Se ha añadido al artículo un capítulo que indica el coste económico de las nuevas mezclas.

(3) El resumen está completo y bien estructurado y explica muy bien el contenido del documento. No obstante, la parte relativa a los resultados podría proporcionar indicadores numéricos obtenidos en la investigación.

Respuesta: Se ha tenido en cuenta el comentario.

Capítulo 1 Introducción

(4) La novedad del estudio no es lo suficientemente evidente. En la sección de introducción, resalte la contribución de su trabajo colocándolo en contexto con el trabajo que ha realizado anteriormente en el mismo dominio.

Respuesta: La novedad de este estudio se explica en el párrafo de la Tabla 1. Sin embargo, hemos tenido en cuenta este comentario al reescribir el párrafo mencionado anteriormente .

(5)The first paragraph introducing the research topic gives a too simple, and even incomplete, view of the problems related to your topic and should be revised and completed with citations to authority references (Compressive properties of rubber-modified recycled aggregate concrete subjected to elevated temperatures; Experimental and Theoretical Investigation on the Thermo-Mechanical Properties of Recycled Aggregate Concrete Containing Recycled Rubber;Axial compression behavior of recycled-aggregate-concrete-filled GFRP–steel composite tube columns).

Answer: The comment has been taken into account.

(6)On a general level, the study of the proposed detection techniques is reasonable, and the explanation of the objectives of the work may be valid. However, the limitations of your work are not rigorously assumed and justified. 

Respuesta: Este estudio se ha realizado con el máximo rigor, siguiendo las metodologías establecidas por la legislación europea para la caracterización de los materiales y hormigones, así como las estipulaciones sobre hormigón recogidas en los distintos códigos estructurales. Seguir esta metodología permite garantizar la validez de los resultados mostrados en este estudio. Por último, esta metodología se describe en los estándares referenciados principalmente en la Tabla 2 y la Tabla 4.

Capítulo 2: Materiales y métodos

(7) En este párrafo se utiliza una gran cantidad de oraciones largas en inglés, se recomienda dividirlas en algunas sencillas en inglés para que los lectores comprendan mejor.

Respuesta: Se ha tenido en cuenta el comentario.

(8) La parte de superposición de curvas de la Figura 2 es demasiado, y se recomienda cambiar un gráfico claro o un gráfico magnífico para facilitar al lector.

Respuesta: En este sentido, las curvas granulométricas de la fracción gruesa mixta y la fracción natural son las mismas. Es por eso que ambas curvas se superponen.

Capítulo 3: Propiedades en estado fresco

(9) En este capítulo se mencionan una gran cantidad de sustantivos profesionales, y se recomienda a los autores que proporcionen explicaciones adecuadas de algún vocabulario profesional para que los lectores puedan comprender mejor.

Respuesta: Muchas gracias por el comentario. El artículo ha sido traducido por un traductor nativo profesional.

Reviewer 2 Report

  1. The term "eco-design" is mentioned in only one sentence in the article. This is the following sentence:

Regarding the fine fraction of RCA, it reveals that, with the exception of the American continent, its use is allowed in the eco-design of new concretes.

Similar to the previous one, the term "Circular Economy" is mentioned in only one sentence in the article. This is the following sentence:

The use of recycled aggregate from C&DW is one of the most widespread strategies to simultaneously achieve the double goal of instituting the concept of circular economy and sustainability in construction.

There are no eco-design results in the article. There is no discussion from the point of view of the circular economy. Authors must either correct the title of the article or significantly change the structure of the article. For eco-design, a detailed LCA (Life-cycle assessment), cost analysis and the like must be done.

  1. It is similar with keywords. I think they need to be corrected.
  2. The last paragraph of the Introduction section should be rewritten. First, explicitly write the shortcomings of previous similar research. After that, write the goals of your research. Emphasize the scientific benefit of your research.
  3. At the beginning of the "2. Materials and Methods" section, present a flow chart of the applied methodology and describe it. It will be easier to follow the research.
  4. Further elaborate in the article on the universality of your methodology.
  5. Nine mixes are used in the research. Why are these mixtures representative?
  6. The authors state the following: "In order to design and formulate the mixes we used the DOE method [45], ..." How did you use DOE.
  7. Potential errors were not considered. Are there any errors? What is the impact of errors on the obtained results? Elaborate further in the article.
  8. It would be good to do a sensitivity analysis and uncertainty analysis of the results.
  9. In the Conclusions section, only the results already shown are repeated. It should also be emphasized: the scientific benefits of your research, the disadvantages of your research and the directions of future research.

Author Response

Revisor 2:

En primer lugar, me gustaría agradecerles los comentarios realizados. Las modificaciones realizadas se resaltan en verde.

  1. El término "ecodiseño" se menciona en una sola frase del artículo. Esta es la siguiente oración:

Regarding the fine fraction of RCA, it reveals that, with the exception of the American continent, its use is allowed in the eco-design of new concretes.

Similar to the previous one, the term "Circular Economy" is mentioned in only one sentence in the article. This is the following sentence:

The use of recycled aggregate from C&DW is one of the most widespread strategies to simultaneously achieve the double goal of instituting the concept of circular economy and sustainability in construction.

There are no eco-design results in the article. There is no discussion from the point of view of the circular economy. Authors must either correct the title of the article or significantly change the structure of the article. For eco-design, a detailed LCA (Life-cycle assessment), cost analysis and the like must be done.

Answer: The comment has been taken into account.

2. It is similar with keywords. I think they need to be corrected.

Answer: The keywords have been changed.

  1. The last paragraph of the Introduction section should be rewritten. First, explicitly write the shortcomings of previous similar research. After that, write the goals of your research. Emphasize the scientific benefit of your research.

Answer: The last paragraph has been rewritten. Likewise, the previous paragraphs describe the current state of the knowledge, with the penultimate paragraph highlighting existing shortcomings and the novelty of this study. However, if you believe it is necessary to insert any additional comments, please do not hesitate to let us know.  

  1. At the beginning of the "2. Materials and Methods" section, present a flow chart of the applied methodology and describe it. It will be easier to follow the research.

Answer: The comment has been taken into account.

  1. Further elaborate in the article on the universality of your methodology.

Answer: This study has been conducted with the utmost rigour, following the methodologies established by European legislation for the characterisation of the materials and concretes, as well as the stipulations on concrete included in the various structural codes. Following this methodology makes it possible to guarantee the validity of the results shown in this study. Lastly, this methodology is described in the standards referenced mainly in Table 2 and Table 4.

  1. Nine mixes are used in the research. Why are these mixtures representative?

Answer: When planning research, we have to establish a number of concretes that meet our goals. Whether these mixes are representative or not depends on the number of test bodies of a single mix and of the methodology used for its characterization.

Regarding the percentage, Spain currently has a maximum coarse recycled concrete addition percentage of 20% for structural concretes. In light of this limit, as well as the existing international regulatory framework, we established a minimum percentage of 25%, which increases with the individual or simultaneous addition of the coarse or fine fraction of MRA.

  1. The authors state the following: "In order to design and formulate the mixes we used the DOE method [45], ..." How did you use DOE.

The dosage method used was the DOE British Method, detailed in reference 45 of the article. The methodology used to conduct their formulation is the one included in said document. 

  1. Potential errors were not considered. Are there any errors? What is the impact of errors on the obtained results? Elaborate further in the article.

Answer: When processing the results, we verified at all times that the deviations were under the maximum limit allowed by current regulation. This is why the obtained results are considered representative and are valid from a statistical point of view.

  1. It would be good to do a sensitivity analysis and uncertainty analysis of the results.

Answer: Table 3 shows the Density and Compressive Strength with the Deviations, which are a way of assessing measuring errors.

10. En la sección Conclusiones, solo se repiten los resultados ya mostrados. También debe enfatizarse: los beneficios científicos de su investigación, las desventajas de su investigación y las direcciones de la investigación futura.

Respuesta: Se ha tenido en cuenta el comentario.

Reviewer 3 Report

Dear authors,

Thank you very much for preparing the article. In my opinion, the name is unnecessarily long - you should be able to prepare a name that is quite concise and shorter at the same time.

With all due respect, the abstract is very strange and contains errors.

The introductory chapter is processed using a large number of facts and references. Obviously, you are familiar with the topic, but it is necessary to divide it into subchapters because it is so long.
However, I would like to suggest adding a few articles on this topic of recycled concrete from another aggregate:
10.1007/s42947-020-0217-7
10.3390/ma13235501
10.1016/j.jobe.2021.102567

The description of the material of natural and recycled aggregates is very good.
Likewise, the description of the tested samples and the tested methods is correct. However, for Chapter 2.2 and Table 4, a slightly more comprehensive description of the equipment used for each method and a brief description would be appropriate. Not every reader is exactly familiar with the procedure and standards.

I praise that you have prepared 9 different mixtures with the same design strength and w / c ratio.

I can't evaluate the results of individual properties of fresh concrete - but I appreciate that you discuss them with the literature and it corresponds.

The results of hardened concrete are interesting.
I have a question - in the note, to table 7 you write that the ratio between cubic and cylindrical compressive strength is 0.9 - where did you find out and is it true ??

From your results of comparing strength with M1, it follows that recycled concrete is weaker in the first days (up to 90 days) - isn't this an adverse effect? Even though it is a percentage, it is still an interesting effect.

For chapters 3.2.2 and 3.2.3 I recommend supplying graphs of results.

The conclusions correspond to the findings.

A few notes:
- references are definitely poorly prepared - format, content, and editing is bad - the word bibliography has nothing to do there,
- the references contain enough articles from Isabel Fuencisla 5 Sáez del Bosque, with all due respect, even though you have a total of 88 references, 9 articles are enough,
- The formatting of text and tables contains errors and is divided into several pages, which is not good.
- The article contains a large number of language errors and typos.

You have to work significantly on the whole text because your results are very useful, but it is necessary to present them well.

Author Response

Reviewer 3:

Firstly, thank you for the comments made. Modifications performed are highlighted in blue.

Thank you very much for preparing the article. In my opinion, the name is unnecessarily long - you should be able to prepare a name that is quite concise and shorter at the same time.

Answer: The comment has been taken into account.

With all due respect, the abstract is very strange and contains errors.

Answer: I would be grateful if you could point out which aspects I should add or modify.

The introductory chapter is processed using a large number of facts and references. Obviously, you are familiar with the topic, but it is necessary to divide it into subchapters because it is so long.
However, I would like to suggest adding a few articles on this topic of recycled concrete from another aggregate:
10.1007/s42947-020-0217-7
10.3390/ma13235501
10.1016/j.jobe.2021.102567

Answer: 10.1016 / j.jobe.2021.102567 has been added.

The description of the material of natural and recycled aggregates is very good.
Likewise, the description of the tested samples and the tested methods is correct. However, for Chapter 2.2 and Table 4, a slightly more comprehensive description of the equipment used for each method and a brief description would be appropriate. Not every reader is exactly familiar with the procedure and standards.

I praise that you have prepared 9 different mixtures with the same design strength and w / c ratio.

Answer: The equipment used for the physical and mechanical characterisation of the concretes is well known and is included in the referenced regulation. This is why said information has been omitted, as well as to not make chapter 2 too lengthy.

I can't evaluate the results of individual properties of fresh concrete - but I appreciate that you discuss them with the literature and it corresponds.

Answer: I do not understand this comment. The properties of fresh concrete are shown in Table 6.

I can't evaluate the results of individual properties of fresh concrete - but I appreciate that you discuss them with the literature and it corresponds.

Answer: There is consensus on an international level that a factor of 0.9 must be used to transform the results of compressive strength of 15x15x15 cm cubic test bodies into 15x30 cm cylindrical test bodies. This figure is established in the Spanish Code on Structural Concrete.

From your results of comparing strength with M1, it follows that recycled concrete is weaker in the first days (up to 90 days) - isn't this an adverse effect? Even though it is a percentage, it is still an interesting effect.

Answer: In this regard, the performance loss due to the addition of recycled aggregate does not represent an adverse effect, as it is always under the percentage of replacement. Lastly, compressive strength after seven days is greater than 25 MPa. 

For chapters 3.2.2 and 3.2.3 I recommend supplying graphs of results.

Answer: As the results for all mixes are similar, a graph will not be able to show these differences. This is why we chose to show them in tables, to make them easier to understand by readers.

The conclusions correspond to the findings.

A few notes:

- references are definitely poorly prepared - format, content, and editing is bad - the word bibliography has nothing to do there,

Answers: The references have been modified.

- the references contain enough articles from Isabel Fuencisla 5 Sáez del Bosque, with all due respect, even though you have a total of 88 references, 9 articles are enough,

Answer: All references used show a conclusion or an interesting piece of data for this study, which is why none have been removed.

- The formatting of text and tables contains errors and is divided into several pages, which is not good.

Answer: These tables will not appear in the final version, as the layout will be handled by the publisher, who will present it properly.

- The article contains a large number of language errors and typos.

Answer: The article was translated by an official translator.

You have to work significantly on the whole text because your results are very useful, but it is necessary to present them well.

Answer: The article was translated by an official translator.

Round 2

Reviewer 2 Report

The article has been updated.

Author Response

The authors wish to thank the reviewer 2 for the excellent review work. No doubt his/her critical and constructive analysis has contributed to a substantial improvement in this manuscript. Thank you very much.

Reviewer 3 Report

Dear authors,
With all due respect, your reactions are not always pleasant and it is not professional.
I made a great effort to move your article further in the required quality.
If you feel so confident that you can ignore many of my comments, references and ideas, then have the article published and think about yourself.
Regards,

Author Response

Reviewer 3

The authors wish to thank the reviewer 3 for the excellent reviewer work as well as the time spent.  No doubt his/her critical analysis and constructive analysis has contributed to a substantial improvement of the manuscript. Changes in the new version of the manuscript are marked in red color. Thanks very much.

References proposed has been added in the manuscript:

10.1007/s42947-020-0217-7

10.3390/ma13235501

10.1016/j.jobe.2021.102567

Some articles of Isabel Fuencisla  Sáez del Bosque has been removed. 

The authors  have responded with humility to the rest of the considerations of the Reviewer 3. We regret that some of their comments have not been understood or we consider that they are not applicable in this study. In any case, we are willing with total humility to resolve, to the best of our ability and knowledge, the suggestions and recommendations of the Reviewer 3.
